# Infant Vitamin D Supplements, Fecal Microbiota and Their Metabolites at 3 Months of Age in the CHILD Study Cohort

**DOI:** 10.3390/biom13020200

**Published:** 2023-01-19

**Authors:** Xin Zhao, Sarah L. Bridgman, Kelsea M. Drall, Hein M. Tun, Piush J. Mandhane, Theo J. Moraes, Elinor Simons, Stuart E. Turvey, Padmaja Subbarao, James A. Scott, Anita L. Kozyrskyj

**Affiliations:** 1Department of Pediatrics, University of Alberta, Edmonton, AB T6G 1C9, Canada; 2The Jockey Club School of Public Health and Primary Care, The Chinese University of Hong Kong, Hong Kong, China; 3Department of Pediatrics, Hospital for Sick Children, University of Toronto, Toronto, ON M5G 1X8, Canada; 4Department of Pediatrics & Child Health, Children’s Hospital Research Institute of Manitoba, University of Manitoba, Winnipeg, MB R3A 1S1, Canada; 5Department of Pediatrics, BC Children’s Hospital, University of British Columbia, Vancouver, BC V6H 0B3, Canada; 6Dalla Lana School of Public Health, University of Toronto, Toronto, ON M5T 3M7, Canada; 7Department of Pediatrics, University of Alberta, 3-527 Edmonton Clinic Health Academy, 11405–87 Avenue, Edmonton, AB T6G 1C9, Canada

**Keywords:** Cholecalciferol, glycerol, 1,2-propanediol, microbiome, breastfeeding, supplementation

## Abstract

Infant vitamin D liquid formulations often contain non-medicinal excipients such as glycerin (ie. glycerol) and 1,2-propanediol (1,2-PD). We examined whether infant vitamin D supplementation is associated with fecal glycerol and 1,2-PD concentrations at 3 months of age and characterized associations between these two molecules, and gut microbiota and their metabolites. Fecal metabolites and microbiota were quantified using Nuclear Magnetic Resonance Spectroscopy and 16S rRNA sequencing, respectively, in 575 infants from the CHILD Study at 3 months of age. Vitamin D supplement use was determined using questionnaires. Vitamin D supplementation was associated with greater odds of high 1,2-PD (adjusted OR 1.65 95% CI: 1.06, 2.53) and with decreased odds of high fecal glycerol (adjusted OR: 0.62 95% CI: 0.42, 0.90) after adjustment for breastfeeding and other covariates. Our findings were confirmed in linear regression models; vitamin D supplementation was positively associated with fecal 1,2-PD and inversely associated with glycerol (aβ: 0.37, 95% CI 0.03, 0.71 & aβ: −0.23 95% CI −0.44, −0.03, respectively). Fecal 1,2-PD and glycerol concentrations were negatively correlated with each other. Positive correlations between fecal 1,2-PD, *Bifidobacteriaceae*, *Lactobacillaceae*, *Enterobacteriaceae* and acetate levels were observed. Our research demonstrates that infant vitamin D supplement administration may differentially and independently influence infant gut microbiota metabolites.

## 1. Introduction

Vitamin D is an important micronutrient for infant bone growth and immune system development [1,2], and has been shown to protect against respiratory infection, including COVID-19 infection [3,4]. The current recommendation in Canada is that all exclusively breastfed, healthy, term infants receive 10 µg/day (400 IU/day) of vitamin D until the infant’s diet contains at least this amount of vitamin D from other sources [5]. Consequently, 75% of Canadian breastfed infants receive vitamin D drops [6]. Vitamin D supplements are administered to infants as liquid formulations that often contain non-medicinal excipients such as glycerin (i.e., glycerol) and 1,2-propanediol (1,2-PD) [7]. As the structural backbone of all lipid compounds, glycerol is also a component of lipid-based solvents found in some vitamin D supplements. Gut microbiota have a strong capacity to rapidly metabolize glycerol to 1,3-propanediol; they can also convert glycerol to 1,2-PD in the production of short-chain fatty acids [8,9]. The gut microbiota of breastfed infants supplemented with vitamin D show reductions to species of the *Veillonellaceae* [10,11,12,13], similar to those seen in supplemented adults or adults with higher vitamin D levels [14]. Vitamin D supplementation of adults is associated with functional changes to gut microbiota related to lipid metabolism [15], which may be propagated by non-medicinal excipients of liquid vitamin D formulations. To date, fecal levels of glycerol and 1,2-PD have not been characterized in infants.

We previously reported associations between vitamin D supplementation and gut microbiota profiles in young infants [10]. The aim of the current study was to determine whether infant vitamin D supplementation is associated with fecal glycerol and 1,2-PD concentrations in infants at 3 months of age, independent of confounding factors including breastfeeding status. A secondary objective was to characterize associations between these two molecules, and gut microbiota and their metabolites. These objectives were achieved by determining concentrations of glycerol and 1,2-PD in 575 fecal samples of 3-month-old infants according to vitamin D supplementation status.

## 2. Materials and Methods

### 2.1. Study Population and Design

This study involved a sub-set of 575 infants, whose data and samples were collected as part of the CHILD (Canadian Healthy Infant Longitudinal Development) Cohort Study (www.childstudy.ca accessed on 1 September 2021). This is a multi-ethnic longitudinal general population birth cohort of mother–infant pairs recruited from four sites across Canada (Vancouver, Edmonton, Winnipeg or Toronto) between 2009 and 2012 [16]. Inclusion criteria was singleton birth at ≥35 weeks of gestational age and a birth weight of ≥2500 g, born to pregnant women recruited in the second trimester. In vitro fertilized births were excluded, as were children born with congenital abnormalities or respiratory distress syndrome. The subset for this study was chosen based on the availability of fecal metabolomics data. Covariate data were prospectively collected for all infants and mothers enrolled in the CHILD Study from birth records and through the use of standardized questionnaires. Covariates of interest in this study included: mode of delivery, infant feeding/nutrition, antibiotic use, maternal pre-pregnancy body mass index (BMI), maternal vitamin use, socioeconomic status and age of stool sample collection. Questionnaires regarding infant supplementation were administered to mothers at 3 months postpartum, where mothers indicated all supplements their infant was receiving up to that point, including the type of vitamin (i.e., vitamin D) given. Mothers provided informed consent upon enrollment and the Human Research Ethics Board at the University of Alberta approved this study (#Pro00010073).

### 2.2. Fecal Metabolites and Microbiota

Fecal samples were collected at approximately 3 months of age (mean 3.7 months, SD 1.2) using a standardized protocol during a planned home visit. Samples were refrigerated immediately after collection and during transport and stored at −80 °C until analysis. Methods of sample collection, DNA extraction and amplification, sequencing of 16S ribosomal RNA gene amplicons and characterization of microbial diversity and taxon relative abundance from Illumina (San Diego, CA, USA) in this cohort have been previously described in detail [17].

Infant samples were analyzed for fecal metabolites at The Metabolomics Innovation Centre (TMIC) at the University of Alberta using nuclear magnetic resonance spectroscopy (NMR). All ^1^H-NMR spectra were collected on a 700 MHz Avance III (Bruker) spectrometer equipped with a 5 mm HCN Z-gradient pulsed-field gradient (PFG) cryoprobe. ^1^H-NMR spectra were processed and profiled using a Chenomx NMR Suite Professional software package version 8.1 (Chenomx Inc., Edmonton, AB, Canada). Metabolites were quantified as μmol per gram of feces. Detailed methods of the NMR analysis in this cohort have been previously described [18].

### 2.3. Statistical Analysis

The concentration of fecal metabolites (1,2-PD and glycerol in micromoles per gram feces) was analyzed in original and transformed units, and as a binary variable with the median concentration as the cut-off value. Fisher’s exact tests were run to compare metabolite (above and below the median) concentrations between infants supplemented with vitamin D and those that were not supplemented. Logistic regression modelling was used to determine the association between infant vitamin D supplementation on the concentration (above or below the median) of glycerol and 1,2-PD in infants at 3 months. Linear regression was performed after data transformation and normalization (ln(+X)) to confirm the association between infant vitamin D supplementation and the concentration of glycerol or 1,2-PD in infants. A DAG (directed acyclic graph) was produced based on existing knowledge to inform a minimum set of covariates that should be adjusted for our models (Appendix A). Although not identified in the minimum set, infant age at fecal sampling was also included due to the variation in sampling time. The 15% change in primary estimate rule was then followed to arrive at the final covariates for adjustment. There was a small amount of missing data for some covariates (details are provided in Table 1). Missing data was excluded from the analysis.

Spearman correlation was used to determine the correlation between gut microbiota at the family level, glycerol and 1,2-PD concentrations. The Spearman ranked order analysis was performed using the Hmisc R package (https://hbiostat.org/R/Hmisc/ Accessed on 1 September 2021), followed by visualization using the gplots R package (https://github.com/talgalili/gplots Accessed 1 September 2021). The scatterplot to explore the relationship between taxa of particular interest and measured levels of fecal glycerol and 1,2-PD was made using the ggpubr R package (https://cran.r-project.org/web/packages/ggpubr/index.html Accessed 1 September 2021). Analyses were conducted using STATA release 14 (StataCorp) and R (version 4.1.1)

## 3. Results

Our analysis included 575 infants whose fecal samples were analyzed at approximately 3 months of age, the peak age of exclusive breastfeeding in Canada. Overall, 60% of the sampled infants were supplemented with vitamin D after birth, similar to the percentage use reported in the larger population [10]. Close to 100% received the brand name Enfamil D-Vi-Sol, which contains 400 IU/mL vitamin D3 and non-medicinal ingredients, glycerin and polysorbate 80. The majority (78%) of exclusively breastfed infants received vitamin D supplements, compared to only 24% of exclusively formula-fed infants (Table 1). The median concentration of 1,2-PD and glycerol was 0.59µmol/g feces (IQR: 0.07–3.33) and 1.13µmol/g feces (IQR: 0.00–3.01), respectively. In our samples, 1,2-PD and glycerol were negatively correlated (Spearman’s r −0.16, *p* < 0.001). Our results showed that a greater proportion of infants who were given vitamin D supplements had high fecal 1,2-PD (above the median) compared to infants not given vitamin D drops (61% vs. 34%, respectively; *p* < 0.001, Table 1). Conversely, infants receiving vitamin D supplements were more likely to have low fecal glycerol compared to those not receiving vitamin D supplements (54% vs. 44% respectively; *p* < 0.03, Table 1). A greater proportion of exclusively breastfed infants had higher than median 1,2-PD and glycerol levels compared to other feeding groups (*p* < 0.001 and *p* < 0.02 respectively, Table 1). Higher than median fecal 1,2-PD level were also associated with maternal education, maternal weight status and timing of introduction of solids, whilst higher fecal glycerol level was associated with maternal education, maternal weight and maternal vitamin use (Table 1). Percentages may not add up to 100.0 due to rounding. *p* values were calculated using Fisher’s exact tests (bolded if *p* < 0.05). The high, low level of fecal 1,2-PD was higher and lower than the median of 0.59 (µmol/g), respectively. The high, low level of fecal glycerol was higher and lower than the median of 1.13 (µmol/g), respectively.

The logistic regression analysis showed that infants receiving vitamin D supplements after birth had greater odds of high 1,2-PD (OR 1.65, 95% CI 1.06, 2.53) but 38% lower odds of high glycerol (OR 0.62, 95% CI 0.42, 0.90) relative to those not receiving vitamin D supplements after adjustment for covariates (Table 2). We confirmed this finding with adjusted linear regression models, which showed that the use of vitamin D supplements was positively associated with fecal 1,2-PD and negatively associated with glycerol (adjusted beta-coefficient—aβ: 0.37, 95% CI 0.03, 0.71 & aβ: −0.23 95% CI −0.44, −0.03, respectively) and that these effects were independent of breastfeeding status and other covariates (Table 2).

To explore vitamin D, glycerol and 1,2-PD pathways, we conducted Spearman correlations with gut microbiota and their main metabolites at 3 months of age stratified by vitamin D status. Statistically significant correlations (*p* < 0.05) are reported in heatmap format for taxon abundance of the 17 top microbiota families (Figure 1) and for lactate and short-chain fatty acid concentrations (Figure 2). Among infants receiving vitamin D drops (Figure 1): i) fecal glycerol was positively correlated with the abundance of *Prevotellaceae* (r = 0.20), *Pasteurellaceae* (r = 0.25) and *Veillonellaceae* (r = 0.16), and ii) fecal 1,2-PD was also positively correlated with the abundance of *Pasteurellaceae* (r = 0.16), as well as of *Lactobacillaceae* (r = 0.19), *Bifidobacteriaceae* (r = 0.17), *Staphylococcaceae* (r = 0.22) and *Enterobacteriaceae* (r = 0.03). The *Lachnospiraceae, Ruminococcaceae* and *Veillonellaceae* were less abundant with increasing fecal levels of 1,2-PD.

In cells Spearman coefficients with *p*-value lower than 0.05 are displayed. The absence of spearman coefficients represents statistical insignificance (*p* > 0.05). At the bottom of the heatmap, “Yes” or “No” indicates whether or not vitamin D supplements were used before 3 months of age. The dendrogram tree to the left side of the heatmap indicates clustering of taxa based on Euclidean distance.

**Figure 2 biomolecules-13-00200-f002:**
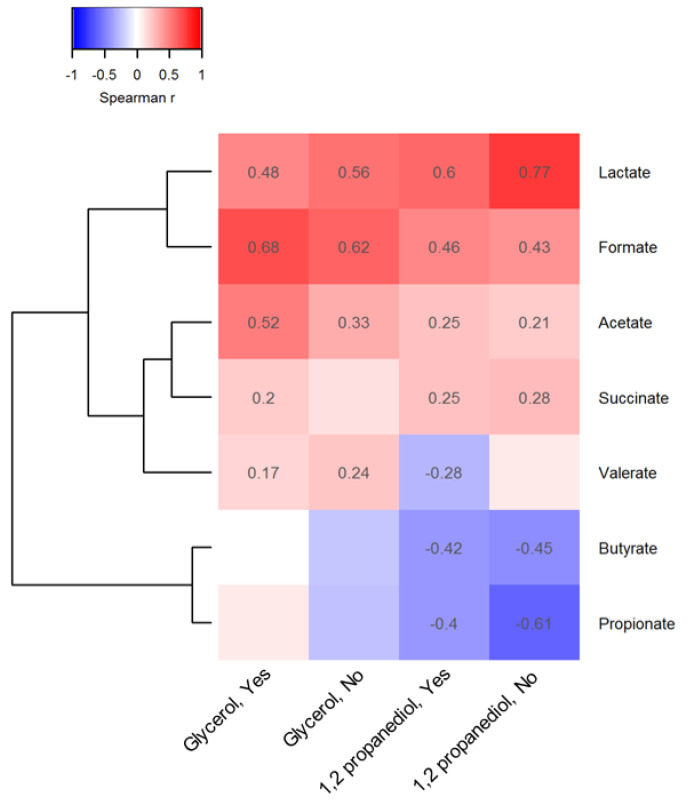
Association between observed 1,2 propanediol and glycerol and representative metabolites.

In cells Spearman coefficients with *p*-value lower than 0.05 are displayed. The absence of spearman coefficients represents statistical insignificance (*p* > 0.05). To the bottom of the heatmap, “Yes” or “No” indicates whether or not vitamin D supplements were used before 3 months of age. The dendrogram tree to the left side of the heatmap indicates clustering of taxa based on Euclidean distance.

Similar 1,2-PD-microbiota correlations were observed in infants not supplemented with vitamin D. Few correlations were observed between glycerol levels and microbial taxon abundance in the stool of the infants not receiving vitamin D supplements. For both compounds, many correlations with microbial metabolites were also independent of vitamin D supplementation but they were of a stronger magnitude compared to microbiota correlations (Figure 2); fecal lactate (r = 0.60), formate (r = 0.46) and acetate (r = 0.25) levels were positively related to 1,2-PD levels when vitamin D supplements were given. However, fecal glycerol levels were more strongly correlated with acetate (r = 0.52) in infants receiving vitamin D supplements versus those who were not. Fecal butyrate and propionate levels declined with increasing 1,2-PD levels in all infants.

## 4. Discussion

In our study of 575 term infants from a general population cohort, we found that vitamin D supplementation of infants was associated with higher fecal 1,2-PD concentrations (aβ: 0.37 [95% CI: 0.03, 0.71]) and lower glycerol concentrations (aβ: −0.23 [95% CI: −0.44, −0.03]), even after adjusting for breastfeeding status. Whilst studies to date, including our own, have reported on the impact of maternal or infant vitamin D supplementation on the infant gut microbiome [10,11], little is known about infant routine vitamin D supplementation and microbial metabolites. Our interest in the 1,2-PD and glycerol (glycerin) metabolites stems from the fact that both are used widely in the pharmaceutical and food industry as additives [9]. In infants, their presence in vitamin D supplements as non-medicinal excipients may become a significant source of substrates for gut microbiota if they are unabsorbed and reach the colon [7]. Commonly-resident gut bacteria in infants also produce these compounds, which are then used in cross-feeding reactions; 1,2-PD is made from breast milk sugars and glycerol, and glycerol from milk lipids [9,19]. Indeed, the extent of exclusive breastfeeding substantially reduced the magnitude of vitamin D’s association with 1,2-PD in our study but it did not negate it, suggesting an additional biological role for vitamin D supplementation in 1,2-PD production. Putatively greater colonic availability of glycerol from glycerin in the D-Vi-Sol vitamin D formulation could promote glycerol conversion by microbiota to 1,2-PD [20].

Finding higher fecal 1,2-PD concentrations in breastfed infants is consistent with the capacity of bifidobacteria, especially as part of a consortium of infant-derived species, to metabolize free fucose or fucose-containing human milk oligosaccharides to 1,2-PD [21,22,23]. Some gut enterobacteria produce 1,2-PD from rhamnose, also found in milk [24]. These pathways increase 1,2-PD availability as a substrate for lactobacilli [25]. Indeed, we observed positive correlations between fecal 1,2-PD, *Bifidobacteriaceae*, *Lactobacillaceae* and *Enterobacteriaceae* levels. Glycerol, a structural component of milk lipids [26], was also more abundant in the stool of exclusively breastfed infants. In contrast to 1,2-PD, fecal glycerol levels were inversely associated with infant vitamin D supplementation and this was independent of breastfeeding. Glycerol is converted to 1,2-PD by *E. coli* or utilized by *Lactobacillus* and other microbiota in acetate production [20,24,25]. In our study, fecal glycerol was inversely related to 1,2-PD levels and more strongly correlated with fecal acetate levels in the presence of vitamin D supplementation. This observation could be the outcome of the aforementioned glycerol-1,2-PD microbiota pathway. Alternatively, milk or formula glycero-lipids may have been used up during intestinal absorption of vitamin D, [27] and, separately, 1,2-PD was produced by microbiota from milk or formula sugars further down in the colon [9].

Since there is significant heterogeneity in serum vitamin D levels with the oral supplementation of infants [28], it is conceivable that unabsorbed vitamin D and its formulation excipients reach the colon to influence gut microbial composition. Indeed, pre–post vitamin D supplementation studies show non-serum response (ie. lower vitamin D levels due to poor absorption) in adults to be accompanied with increases to gut *Bifidobacterium* [15]; as we noted, bifidobacteria have the capacity to break down milk sugars to produce 1,2-PD and initiate cross-feeding by lactobacilli and other microbiota to produce acetate or other short-chain fatty acids [24]. Probiotic and vitamin D co-administration trials suggest an additional benefit of vitamin D supplementation—a reduction in colic and intestinal inflammation [29], which could operate through 1,2-PD and short-chain fatty acid generation. Vitamin D treatment of adults with *C. difficile* disease has also resulted in fecal enrichment of bifidobacteria and other microbiota during the recovery phase [30]. Further, since both vitamin D and gut microbiota have been associated with reductions to allergic disease development [2], these pathways may also include the 1,2-PD metabolite. One of these pathways could involve B cell activation, which has been shown in animal model and in vitro studies following the incubation of B cells with 1,2-PD [31].

This is the first examination of non-medicinal excipients in a common medication formulation given to young infants that could be utilized as substrates by gut microbiota. Little is known about the role of glycerol and 1,2-PD in gut microbiota metabolism, and only recently has evidence for glycerol metabolism to 1,2-PD been published [20]. Our human infant study is based on a well-characterized, population-based cohort study with adequate sample size to measure fecal levels of these two compounds and adjust for important confounding factors, in particular breastfeeding. On the other hand, this study has some limitations, including cross-sectional determination of fecal metabolites and microbiota and self-reported questionnaire sources for infant vitamin D supplementation administration. The majority of infants were given D-Vi-Sol, which also contains the non-medicinal emulsifier, polysorbate-80. We did not measure this fecal metabolite, which has been shown to have varying effects on the gut microbiome [32]. Other infant medications may also contain excipients. However, vitamin D supplements are by far the most common medication given to infants [33], and our studied fecal metabolite levels were much higher in breastfed than non-breastfed infants, which is consistent with vitamin D supplement recommendations.

In conclusion, in accordance with public health recommendations, the majority of exclusively breastfed infants are receiving vitamin D drops. We found that fecal 1,2-PD levels were positively associated and fecal glycerol levels inversely associated with the use of vitamin D drops in infants, independent of breastfeeding status. Our research has demonstrated that infant vitamin D supplement administration can differentially and independently influence infant gut microbiota metabolites.

## Figures and Tables

**Figure 1 biomolecules-13-00200-f001:**
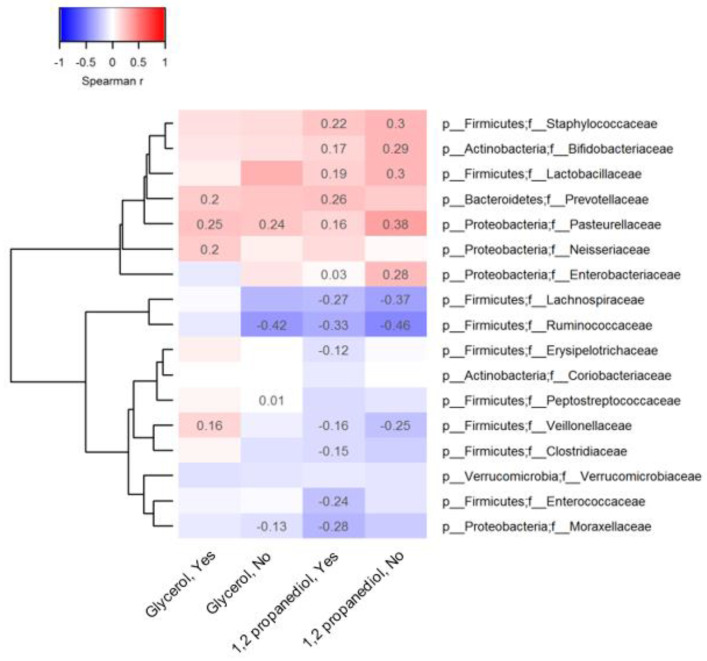
Association between observed 1,2 propanediol, glycerol and representative taxa at the family level.

**Table 1 biomolecules-13-00200-t001:** Characteristics of the analytic samples according to vitamin D drop usage and fecal 1,2-PD and glycerol levels (*n* = 575).

Characteristics	Infant Vitamin D Drop Use, No. (%)		*p*	Fecal 1,2-PD Levels, No. (%)		*p*	Fecal Glycerol Levels, No. (%)		*p*
	No	Yes		Low	High		Low	High	
Infant vitamin D									
Yes				132 (38.8)	208 (61.2)	**<0.001**	183 (53.8)	157 (46.2)	0.02
No				155 (66.0)	80 (34.0)		104 (44.3)	131 (55.7)	
Birth mode (*n* = 557)			0.455			0.720			0.1
Vaginal no IAP	132 (42.9)	176 (57.1)		149 (48.4)	159 (51.6)		144 (46.8)	164 (53.3)	
Vaginal with IAP	39 (34.8)	73 (65.2)		57 (50.9)	55 (49.1)		57 (50.9)	55 (49.1)	
Elective CS	21 (43.8)	27 (56.3)		25 (52.1)	23 (47.9)		24 (50.0)	24 (50.0)	
Emergency CS	34 (36.2)	55 (61.8)		49 (55.1)	40 (44.9)		55 (61.8)	34 (38.2)	
Feeding modes (*n* = 572)			**<0.001**			**<0.001**			**0.02**
Exclusive breastfeeding	50 (22.4)	173 (77.6)		60 (26.9)	163 (73.1)		100 (44.8)	123 (55.2)	
Partial breastfeeding	55 (31.1)	122 (68.9)		83 (46.9)	94 (53.1)		104 (58.8)	73 (41.2)	
Exclusive formula	130 (75.6)	42 (24.4)		143 (83.1)	29 (16.9)		82 (47.7)	90 (52.3)	
Mother’s education status (*n* = 554)			**<0.001**			**<0.001**			**<0.01**
Less than high school	52 (58.4)	37 (41.6)		57 (64.0)	32 (36.0)		39 (43.8)	50 (56.2)	
High school	92 (47.7)	101 (52.3)		108 (56.0)	85 (46.0)		84 (43.5)	109 (56.5)	
College or University	82 (30.2)	190 (69.9)		110 (40.4)	162 (59.6)		158 (58.1)	114 (41.9)	
Solids given at 3 months (*n* = 542)			0.219			**<0.01**			0.689
No	207 (40.1)	309 (59.9)		250 (48.5)	266 (51.6)		265 (51.4)	251(48.6)	
Yes	14 (53.9)	12 (46.2)		20 (76.9)	6 (23.1)		12 (46.2)	14 (53.9)	
Infant antibiotics			0.86			>0.99			0.33
No	228 (40.9)	329 (59.1)		278 (49.9)	279 (50.1)		276 (49.6)	281 (50.5)	
Yes	7 (38.9)	11 (61.1)		9(50.0)	9 (50.0)		11 (61.1)	7 (38.9)	
Maternal pre-pregnancy weight (*n* = 540)			**0.03**			**<0.01**			**0.01**
Underweight	3 (23.1)	10 (76.9)		4 (30.8)	9 (69.2)		10 (76.9)	3 (23.1)	
Normal	105 (36.5)	183 (63.5)		133 (46.2)	155 (53.8)		132 (45.8)	156 (54.2)	
Overweight	55 (45.1)	67 (54.9)		54 (44.3)	68 (55.7)		74 (60.7)	48 (39.3)	
Obese	59 (50.4)	58 (49.6)		73 (62.4)	44 (37.6)		60 (51.3)	57 (48.7)	
Maternal postnatal vitamins (*n* = 438)			0.12			>0.99			**0.05**
No	20 (51.3)	19 (48.7)		19 (48.7)	20 (51.3)		15 (38.5)	24 (61.5)	
Yes	154 (38.6)	245 (61.4)		192 (48.2)	207 (51.8)		221 (55.4)	178 (44.6)	

**Table 2 biomolecules-13-00200-t002:** Regression results for prediction of fecal 1,2-PD, glycerol levels from infant use of vitamin D drops.

Infant Vitamin D	Fecal 1,2-PD (High vs. Low)		Fecal Glycerol (High vs. Low)	
	Crude OR	Adjusted OR ^a^	Crude OR	Adjusted OR ^b^
No (ref.)	-	-	-	-
Yes	3.03 (2.16, 4.35)	1.65 (1.06, 2.53)	0.68 (0.49, 0.95)	0.62 (0.42, 0.90)
	Fecal 1,2-PD		Fecal glycerol	
	Crude β	Adjusted β ^c^	Crude β	Adjusted β ^d^
No (ref.)	-	-	-	-
Yes	1.09 (0.77, 1.42)	0.37 (0.03, 0.71)	−0.18 (−0.35, −0.01)	−0.23 (−0.44, −0.03)

**In** logistic regression models fecal 1,2-PD was dichotomized into high and low levels using the median (0.59 μmol/g) as the cut-off. Fecal glycerol was dichotomized into high and low levels using the median (1.13 μmol/g) as the cut-off. **In** linear regression models, fecal 1,2-PD was transformed by ln (1,2-PD + 0.030667) and fecal glycerol by ln (Glycerol + 0.4052815). ^a^ Adjusted for feeding mode, introduction of solids at 3 months and age of stool collection. ^b^ Adjusted for feeding mode. ^c^ Adjusted for feeding mode, introduction of solids at 3 months and age of stool collection. ^d^ Adjusted for feeding mode, introduction of solids at 3 months and maternal education.

## Data Availability

Restrictions apply to the availability of these data. Data was obtained from the CHILD Cohort Study and are available via childcohort.ca (accessed on 1 September 2021) with the permission of Anita Kozyrskyj and the Child Cohort Study National Coordinating Centre.

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
