# Peer review of "Infant Vitamin D Supplements, Fecal Microbiota and Their Metabolites at 3 Months of Age in the CHILD Study Cohort"

_biomolecules, 2023, doi:10.3390/biom13020200_

Round 1

Reviewer 1 Report

The authors evaluated whether infant vitamin D supplementation is associated with fecal glycerol and 1,2-PD concentrations at 3 months of age. Likewise, the authors determined associations between these two molecules and gut microbiota and their metabolites (lactate and short chain fatty acids). The theme of the manuscript is interesting in the field of public health and contributes to the knowledge about the impact of the administration of dietary supplements on intestinal microbiota and its metabolism. The methodology used is adequate for the objective proposed. However, the authors must describe in detail the methodology used for the determination of the different fecal metabolites and the fecal microbiota. The results were analyzed using statistical tools and presented in tables.  The discussion is adequate and supported by mostly current cited references. I suggest the publication of the manuscript in Biomolecules after minor revision

Some suggestions: 

 Materials and Methods section: 

 The authors must add a subtitle to describe the methodology related to the treatment of fecal samples for intestinal microbiota studies and their determination. Likewise, the authors must add another subtitle to write how fecal metabolites are determined (glycerol, 1,2-PD, lactate and short-chain fatty acids). In the summary section, it is mentioned that they were determined by Nuclear Magnetic Resonance Spectroscopy.

 Figures: In the legend of Figure 1, the authors should describe that "yes" or "No" indicates the use or not of the Vitamin D supplement.

Author Response

Some suggestions: 

Materials and Methods section: 

The authors must add a subtitle to describe the methodology related to the treatment of fecal samples for intestinal microbiota studies and their determination. Likewise, the authors must add another subtitle to write how fecal metabolites are determined (glycerol, 1,2-PD, lactate and short-chain fatty acids). In the summary section, it is mentioned that they were determined by Nuclear Magnetic Resonance Spectroscopy.

Completed as requested.

Figures: In the legend of Figure 1, the authors should describe that "yes" or "No" indicates the use or not of the Vitamin D supplement.

Completed as requested. Thank you for your suggestions.

Reviewer 2 Report

This manuscript uses fecal microbiota and metabolomics in children at approximately 3 months of age to study whether infant vitamin D supplementation is correlated with fecal glycerol and 1,2-PD concentrations. This is a very interesting study; however, I have some serious concerns regarding the introduction and discussion.

1) Introduction, I think authors should add more information about relationships between vitamin D and the gut microbiome or metabolites.

2) Materrials and Methods, the relationship between microbes and metabolites has been a tricky subject to analyze. I suggested authors could use the MetOrigin (Yu, Gang, Cuifang Xu,Danni Zhang, Feng Ju, and Yan Ni. 2022.“MetOrigin: Discriminating the Origins of Microbial Metabolites for Integrative Analysis ofthe Gut Microbiome and Metabolome.”iMeta. e10. https://doi.org/10.1002/imt2.10) or other methods to analysis the relationships gut microbiota and their metabolites based on the biological results.

3)Discussion, authors still did not clarify what biological principles could be explained by the analysis results. It is suggested that authors further discuss the results.

Author Response

1) Introduction, I think authors should add more information about relationships between vitamin D and the gut microbiome or metabolites.

Completed as requested.

2) Materrials and Methods, the relationship between microbes and metabolites has been a tricky subject to analyze. I suggested authors could use the MetOrigin (Yu, Gang, Cuifang Xu,Danni Zhang, Feng Ju, and Yan Ni. 2022.“MetOrigin: Discriminating the Origins of Microbial Metabolites for Integrative Analysis ofthe Gut Microbiome and Metabolome.”iMeta. e10. https://doi.org/10.1002/imt2.10) or other methods to analysis the relationships gut microbiota and their metabolites based on the biological results.

Thank you for recommending the MetOrigin program. We were excited to test it out. As indicated in our paper, the program listed 1,2-PD as of microbial origin and glycerol as generated from co-metabolism of the microbe and host. Unfortunately, no pathways were generated by MetOrigin for 1,2-PD and none of the 3 listed pathways for glycerol included 1,2-PD. This is not surprising as evidence for the glycerol -> 1,2-PD is recent, as per this paper that we originally cited: Clomburg JM, Cintolesi A, Gonzalez R In silico and in vivo analyses reveal key metabolic pathways enabling the fermentative utilization of glycerol in Escherichia coli. Microb Biotechnol 2022;15(1):289-304.Thus, unfortunately, the MetOrigin program was of little help to our paper that reports on 2 under-researched microbiota metabolites.

3)Discussion, authors still did not clarify what biological principles could be explained by the analysis results. It is suggested that authors further discuss the results.

We are not 100% certain on the meaning of ‘biological principles.’ If it is in reference to microbial pathways for the metabolites studied, glycerol and 1,2-PD, the majority of the Discussion speculated on the origins and putative pathways of these 2 metabolites. Perhaps awkward wording interfered with intended messages. As such, we have substantially edited the 3rd paragraph to improved clarity on our speculations of microbiota pathways and health benefits. We have also added content on B cell activation by 1,2-PD as a pathway in reducing allergic disease.

Round 2

Reviewer 2 Report

I had read through the paper titled "Infant vitamin D supplements, fecal microbiota and their metabolites at 3 months of age in the CHILD Study cohort " by Zhao et al. and had no other suggestions. I recommend that this manuscript be accepted.